# Direct observation of mode-specific phonon-band gap coupling in methylammonium lead halide perovskites

Heejae Kim[1], Johannes Hunger [1], Enrique Cánovas[1], Melike Karakus[1], Zoltán Mics[1], Maksim Grechko[1], Dmitry Turchinovich [1], Sapun H. Parekh [1] & Mischa Bonn[1]

Methylammonium lead iodide perovskite is an outstanding semiconductor for photovoltaics. One of its intriguing peculiarities is that the band gap of this perovskite increases with increasing lattice temperature. Despite the presence of various thermally accessible phonon modes in this soft material, the understanding of how precisely these phonons affect macroscopic material properties and lead to the peculiar temperature dependence of the band gap has remained elusive. Here, we report a strong coupling of a single phonon mode at the frequency of ~1 THz to the optical band gap by monitoring the transient band edge absorption after ultrafast resonant THz phonon excitation. Excitation of the 1 THz phonon causes a blue shift of the band gap over the temperature range of 185 ~ 300 K. Our results uncover the mode-specific coupling between one phonon and the optical properties, which contributes to the temperature dependence of the gap in the tetragonal phase.

[1] Department of Molecular Spectroscopy, Max Planck Institute for Polymer Research, Ackermannweg 10, 55128 Mainz, Germany. Correspondence and requests for materials should be addressed to H.K. (email: kim@mpip-mainz.mpg.de)

Methylammonium lead iodide (MAPbI₃) perovskite has emerged as an outstanding material for photovoltaic applications[1], combining remarkable power conversion efficiency[2, 3] with cost effective fabrication protocols[4]. Its excellent performance in photovoltaic applications results from such properties as high-absorption coefficients in the visible range[5], lowexciton-binding energies[6], longcharge carrier diffusion lengths[7], and low mid-gap trap densities[8]. One more peculiarity of MAPbI₃ perovskite is its soft nature: the ions in the crystallographic unit cell reside in shallow minima of the potential energy landscape[9–11]. The implications of the soft nature of this material on the photovoltaic power conversion efficiency has been discussed controversially in the literature[12–17]. While the large variability of the atomic coordinates within the unit cell (i.e., structural disorder) has been suggested to enhance carrier diffusion lengths[12–14], it also results in a broad distribution of optical properties (i.e., electronic disorder), which may lead to energetic losses in a photovoltaic process[15–17].

This soft material has thermally accessible phonon modes (thermal energy at room temperature is equivalent to ~ 6 THz)[18–22], which naturally displaces atoms from their equilibrium positions. Since the electronic structure is determined by the precise atomic positions in any material, the position of the conduction and valence bands in a defect-free semiconductor is governed by phonon populations at a finite temperature. Thus, the average phonon energy and average phonon anharmonicity (responsible for thermal expansion) are generally sufficient to describe the temperature-dependent band gap shift of conventional semiconductors (e.g., Si, GaP, and GaAs): a well-known decrease of the band gap with increasing temperature[23, 24]. Intriguingly, for MAPbI₃ perovskite the band gap increases with lattice temperature[25, 26]. A recent theoretical study has also reported the positive correlation between thermal expansion and the band gap[26]. However, both the thermal expansion and the band gap variations are caused by the complex, temperature-dependent behavior of phonons. Additionally, the thermal population of phonons in MAPbI₃ perovskites is expected to dominate the electronic disorder, as the contribution of lattice displacements outweighs defect or impurity-related contributions to the lineshape of photoluminescence[15, 27]. Despite the importance of phonons for the opto-electronic properties, mode-specific contributions of these phonons to the optical properties of MAPbI₃ have remained elusive. To understand the peculiar thermal dependence of the opto-electronic behavior of MAPbI₃, it is, therefore, crucial to identify which and how specific phonon modes affect the optical band gap of this technologically relevant material.

In this work, we explore the coupling between a specific phonon mode and the optical band gap in the MAPbI₃ perovskite. For MAPbI₃, the phonon modes up to 5 THz are dominated by the motion of the inorganic Pb and I ions[18–22]. Since the conduction and valence bands of MAPbI₃ perovskites largely consist of Pb 6p and I 5p orbitals[28, 29], the band gap is expected to be specifically sensitive to these lowest phonon modes. Indeed, theoretical studies of geometrically constrained MAPbI₃ perovskites[30] have predicted that changes in the Pb-I sublattice (e.g., the Pb-I-Pb angle) affect the position of the band gap. Here, we experimentally study the coupling of these particular phonons related to Pb-I-Pb angle with the optical band gap. We use intense THz pulses for direct quasi-resonant excitation of these low-

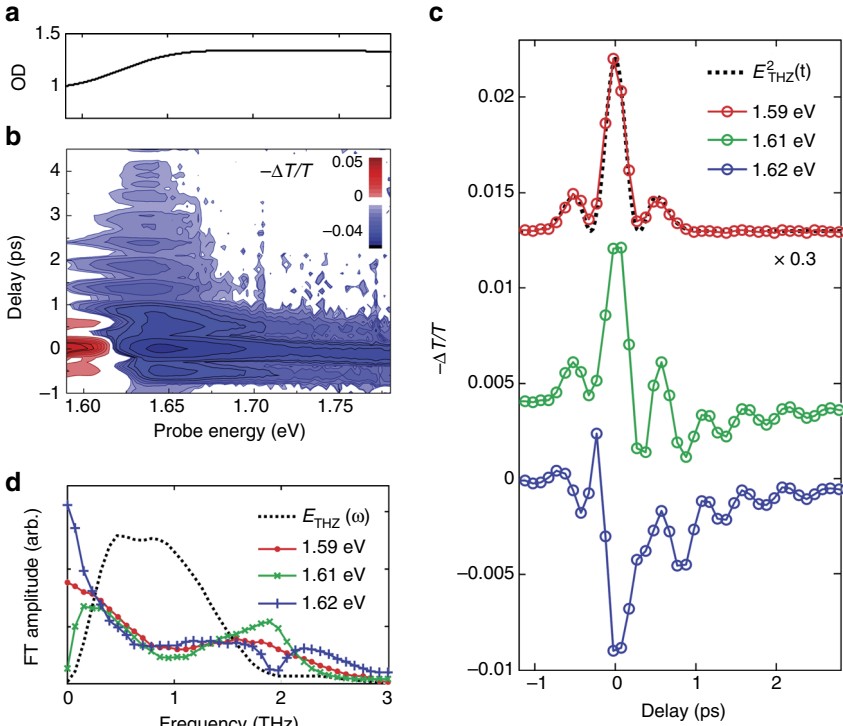

**Fig. 1** Experimental optical band gap modulation by phonon excitation. **a** Representative linear optical absorption spectrum. **b** Two-dimensional THz pump —visible probe spectra of the as-prepared MAPbI₃ polycrystalline film measured at 295 K. The THz-induced differential transmission ($-\Delta T/T$) spectra are recorded as a function of the pump-probe delay time. **c** Time-resolved pump probe signals at probe frequencies of 1.59, 1.61, and 1.62 eV. Traces of 1.61 and 1.62 eV (just above the band gap) have three distinct components—instantaneous (electro-absorption), exponentially decaying, and oscillatory. The trace at 1.59 eV (just below the band gap) shows only an instantaneous response (at $-1 < t < 1$ ps), which scales with the square of the THz field (*dotted black line*). **d** The Fourier transform (FT) of the time traces shows the frequency of the oscillatory component at 1.61 and 1.62 eV by a peak and dip at ~ 2 THz, respectively, compared to that of 1.59 eV (pure electro-absorption and no oscillation). The *dotted black curve* is the FT of the incident THz pulse profile as measured from electro optic sampling in a ZnTe crystal (see SI)

frequency phonon modes, and broadband visible pulses for simultaneous monitoring the optical band gap absorption. In a THz-pumped perovskite, we observe a transient optical band gap shift and a periodic modulation of the gap at room temperature, which points to a direct connection between a targeted phonon and the optical band gap. The observed periodicity of optical response of MAPbI$_3$, and our nonlinear optical response modeling revealed that it is the ~ 1 THz phonon, which has been theoretically assigned to the Pb-I-Pb angular bending vibration[21], that in fact modulates the optical band gap. Quantitative analysis of the phonon-induced shift of the optical band gap shows that this 1 THz phonon mode has a high efficiency in increasing the band gap by thermal population, consistent with the unconventional variation of the optical band gap in MAPbI$_3$ with temperature[25, 26].

## Results

**Optical band gap modulation by phonon excitation.** To specifically excite phonons in thin films of polycrystalline MAPbI$_3$, we focused the THz pulses with the peak electric field of 100 kV/cm, a center frequency of ~ 0.8 THz, and a bandwidth (FWHM) of ~ 1.0 THz onto the perovskite sample (see the Methods section). The THz pulses have frequency components resonant with inorganic sub-lattice vibrational modes in this material[18–22], including the most prominent phonons at frequencies of ~ 1.0 THz and ~ 1.9 THz (in the tetragonal phase)[20]. The THz pulse-induced modulation of the optical band gap of MAPbI$_3$ (1.61 ± 0.03 eV for our sample) was detected through the variation of the

near-gap optical absorption, over a range of 1.6–1.8 eV. The linear optical absorption spectrum is shown in Fig. 1a. We note that the high background below the band gap (Fig. 1a) is due to the surface morphology of the sample[31]; this effect is later cancelled by obtaining differential spectra, with and without the THz pump. The modulated optical band gap was measured as a function of delay time $t$ between THz excitation and a white-light continuum pulse co-propagating with the THz pulse. To gain spectral information, we dispersed the visible detection pulses in a monochromator, detected the spectra on an array camera, and thus recorded the differential transmission spectra ($-\Delta T/T$) induced by the THz pulses for all optical probe photon energies (for further details, see the Methods section).

In Fig. 1b, we show the differential transmission spectra ($-\Delta T/T$) as a function of delay time at probe energies ranging from 1.6 to 1.8 eV. At delay times $-1 < t < 1$ ps, where the THz and visible pulses temporally overlap, the differential spectra exhibit a THz-induced absorption (red) and transparency (blue) at energies below and above 1.62 eV, respectively. We attribute this quasi-instantaneous differential spectrum to the electro-absorptive effect. This distortion of absorption spectra has also been observed in the presence of lower frequency electric fields[32–35], and the change in absorption has been shown to depend quadratically on the applied electric field strength[32–36]. In line with this notion, we find that the temporal profile of the electro-absorptive signal quantitatively scales with the square of the incident THz electric field (Fig. 1c, *top panel*)[32, 34, 36].

At probe photon energies slightly above the band gap onset (at energies of 1.6–1.67 eV), we observe a modulation of the optical

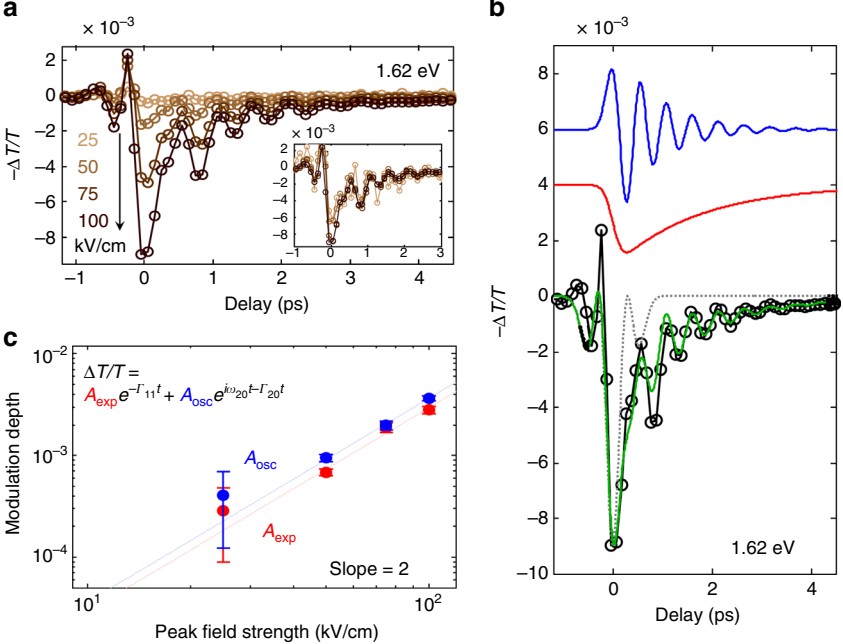

**Fig. 2** Nonlinear optical response to the THz field. **a** Optical transmission changes ($-\Delta T/T$) at probe energy of 1.62 eV induced by THz excitation pulses having peak field strengths of 25, 50, 75, and 100 kV/cm. *Inset*: Each time trace of the optical response is scaled by $\frac{-\Delta T(E_{x,peak})}{T}\left(\frac{E_{0,peak}}{E_{x,peak}}\right)^2$, where $E_{0,peak}$ is the peak field strength (100 kV/cm) of the strongest THz pulse and $E_{x,peak}$ is the peak field strength of a given THz pulse. The overlap of scaled time traces shows the overall quadratic dependence of the transient differential spectra at $t > 1$ ps on the incident THz field strength. **b** Contributions to the optical response ($-\Delta T/T$) as determined from the phenomenological model (damped oscillator, see SI). *Blue* and *red* traces show the decomposed oscillation and exponential components obtained from fitting the model (equation given in **c**) at $t > 1.3$ ps to the data at a probe energy of 1.62 eV (for error estimation, see SI). To extrapolate these contributions to times $t < 1.3$ ps we convolve the modeled response with a Gaussian function (FWHM: 350 fs from the THz pulse width). These contributions together with the instantaneous electro-absorption (*dotted gray line*) agree well with the experimental data (*cf.* modeled *green line* and experimental *black line*). **c** Fitted amplitudes of both oscillatory ($A_{osc}$) and exponential ($A_{exp}$) components (according to the shown equation) scale quadratically on the incident THz peak field strength, reinforcing the second order (nonlinear) nature of the optical response with respect to the THz field

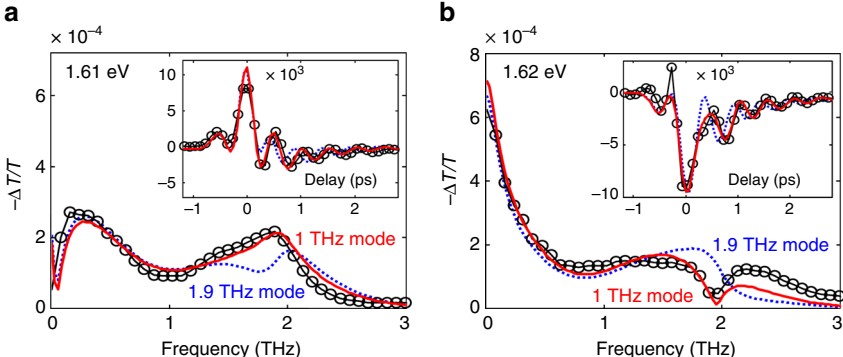

**Fig. 3** Quantitative interpretation of the experiment via the 3rd order nonlinear polarization model. Comparison of the 3rd order nonlinear polarization model describing phonon driven optical band gap modulation (*red solid curve*) with the experimental data (*black circles*) in the frequency domain at the probe frequency of **a** 1.61 and **b** 1.62 eV, respectively. Resonant excitation of 1 THz mode describes the oscillatory behavior very well, whereas the other possibility, non-resonant excitation of 1.9 THz mode shows a 90° phase shift (*blue dashed curve*) for both energies. *Inset*: Same comparison in the time domain. See SI for detailed explanation of the modeling

absorption after the THz pulse passes through (at delay times $t >$ 1 ps). At these probe energies we observe, for several picoseconds, a differential signal in MAPbI$_3$ (a reduced absorption, blue in Fig. 1b), which oscillates with a characteristic period of ~ 0.5 ps. This differential signal at $t > 1$ ps consists of a monotonically decaying and an oscillating contribution, as evident from time traces at several near-gap photon energies (in bottom panels of Fig. 1c). Apart from the quasi-instantaneous electro-absorption, present at all detection energies (*top panel* of Fig. 1c), the two additional components are present in the traces at 1.6–1.7 eV (*dotted gray*, *red*, and *blue curves*, respectively, in Fig. 2b). The differential spectrum indicates a reduced optical absorption induced by THz pump near the band gap (Fig. 1a, b), which corresponds to an effective shift of the optical band gap to higher energies (by up to ~ 0.3 meV, see Supplementary Fig. 1 and Methods section). Even though the optical transition near the band gap in semiconductors can contain contributions from the exciton, the exciton binding energy in this perovskite has been reported to be a few to a few tens of meV, which makes the contribution from the exciton negligible in the room temperature[27].

To obtain further insight into such a modulation of the optical band gap at $t > 1$ ps, we investigate the THz-phonon interaction in detail. Over the delay time range of $t > 1$ ps, the depth of the optical band gap modulation (i.e., the magnitude of the transient spectra) scales with the square of the THz electric field strength, as shown by a series of THz field-dependent measurements (Fig. 2a). In our experimental geometry, the broad band probe pulse serves also as an additional field with the same wavevector as the signal field. In other words, the magnitude of the measured (heterodyne) transient spectra is linearly proportional to the nonlinear polarization of the system. Therefore, the quadratic dependence on the THz field strength (Fig. 2) points to the nonlinear, higher-order nature of the signal, requiring two interactions between the THz electric field and the perovskite.

**Phonon population and coherence**. The occurrence of two light-matter interactions explains the two observed signal contributions: first of all, two interactions with the field results in population in the first excited state of a system (i.e., state $|1\rangle$); Secondly, an oscillating polarization between ground and second excited (overtone) state (i.e., states $|0\rangle$ and $|2\rangle$) is induced. As a result, one can assign the monotonically decaying differential spectra to a shift of the optical band gap to the THz-induced population of a phonon (state $|1\rangle$), and the oscillatory component to a coherent

modulation of the optical band gap at the frequency of the first overtone frequency of the excited phonon.

The monotonically decaying differential spectrum, which reflects a transient blue shift of the optical band gap, can be understood by the transient displacement of a specific lattice coordinate. Population of a vibrationally excited state $|1\rangle$ displaces the lattice coordinate, e.g., the Pb-I-Pb angle in case of ~ 1 THz phonon mode (for the assignment, see the next paragraph). Then, the displacement alters the probability of electronic transitions from state $|1\rangle$ to state $|e\rangle$ (i.e., any electronic excited state) as compared to that of the $|0\rangle \rightarrow |e\rangle$ transition. Thus, the reduced optical absorption near the band gap upon phonon excitation (Fig. 1b) points to a strong coupling between the specific phonon and the band gap (i.e., the electronic states which determine the band gap). Here, the identical transient signals at $t > 1$ ps induced by THz pump pulses with a narrower bandwidth (Supplementary Fig. 2) indicate that phonon modes with a center frequency below ~ 1 THz have no observable contribution to the transient blue shift. Besides, we note that all transient spectra decay to zero at $t > 4$ ps, i.e., the optical band gap returns to its original state (Fig. 1b) without further spectral changes. This means that even though the energy absorbed by the specific phonon (see below) is redistributed over all other available phonon modes (through phonon-phonon coupling)[37, 38], the resulting population of other phonon modes has negligible effect on the electronic transitions near the band gap.

To identify the specific phonon causing the observed optical band gap shift, we investigated the oscillating behavior of the transient differential spectra at $t > 1$ ps. Fourier transformation of the time traces (Fig. 1c, d) provides the frequency of the oscillatory signal and its phase relative to the distinct instantaneous electro-absorption (around $t = 0$, Fig. 1c). By comparing the frequency-domain response recorded at 1.59 eV (*red curve* in Fig. 1c), where only the electro-absorption is present, to the signals at 1.61 and 1.62 eV, where all three contributions are present, we find that the oscillatory component results in a narrow peak (dip at 1.62 eV), at ~ 2 THz (Fig. 1d). The different sign of the signal at ~ 2 THz at 1.61 and 1.62 eV originates from the different sign of the electro-absorption. As discussed above, two THz-sample interactions (Fig. 2c) lead to a wave packet oscillating at the overtone frequency of the phonon (i.e., overtone phonon coherence). Thus, the oscillating behavior of the differential spectra at ~ 2 THz reveals the generation of a wave packet of a phonon centered at ~ 1 THz. In turn, the lattice oscillation along this phonon coordinate shifts the coupled optical band gap periodically.

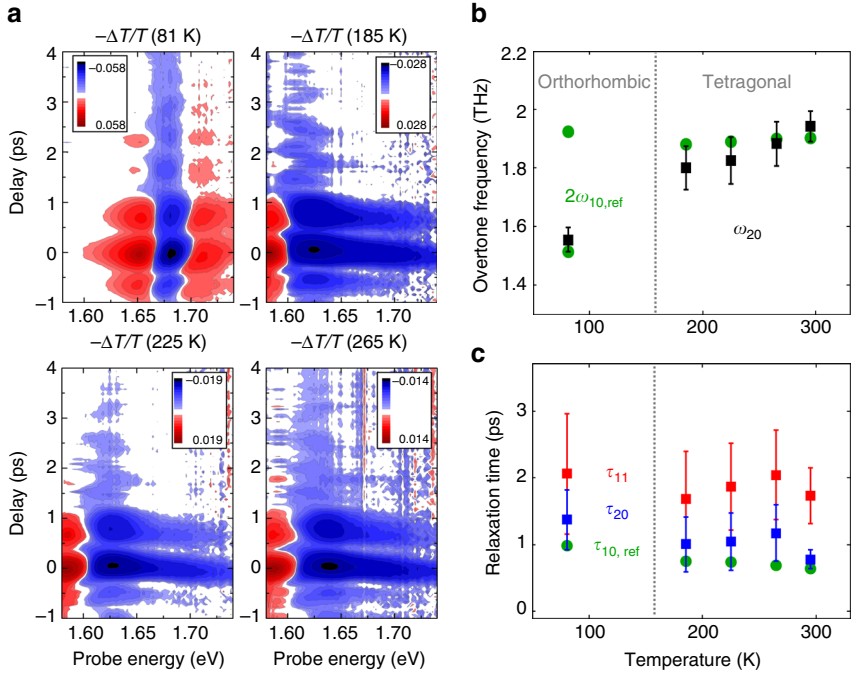

**Fig. 4** Temperature dependence of the phonon-gap coupling. **a** The THz pump visible probe spectra at different temperatures: 81, 185, 225, and 265 K. **b** The temperature dependence of the overtone frequency (*black squares*) obtained by the damped oscillator fit of the data (SI) after the instantaneous response disappears and the reported fundamental dephasing time[20] multiplied by 2 (*green circles*). **c** The temperature dependence of the population relaxation (*red squares*) and the overtone dephasing time (*blue squares*) compared with the reported fundamental dephasing time (*green circles*)[20]. **b**–**c** The *gray dashed line* indicates the temperature where the phase transition between the orthorhombic and the tetragonal structure occurs. The error ranges correspond to the weighted standard deviation of each parameter at different detection energies (for further details see Methods section)

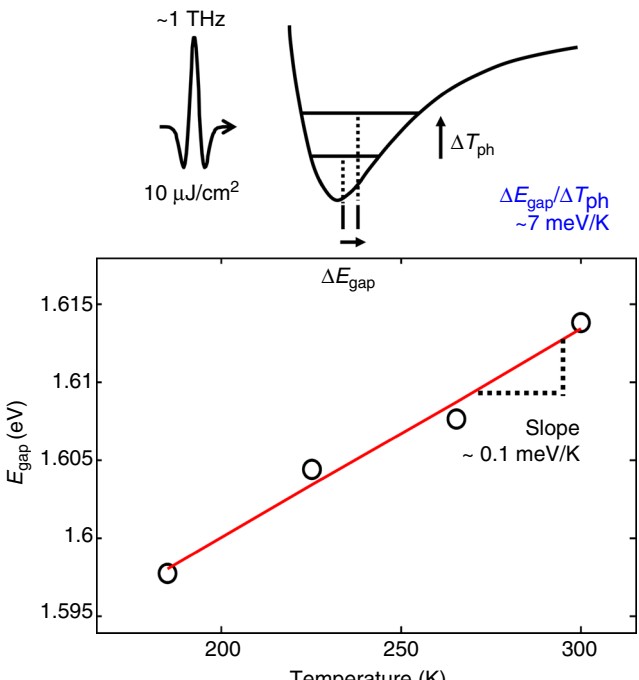

**Fig. 5** Efficiency of the 1 THz phonon in increasing the band gap. The band gap of MAPbI$_3$ varies linearly with temperature at 185–300 K (in the tetragonal phase). Using a linear fit to the data, we extract a variation of the gap by 0.1 meV/K. The population increase of the 1 THz phonon by THz pump (10 μJ/cm$^2$) leads to the blue-shift of the band gap ($\Delta E_{gap}$) by 0.3 meV. The effective phonon temperature ($\Delta T_{ph}$), which is the temperature equivalence of the population increment is estimated to be ~ 0.04 K, which gives higher slope, ~ 7 meV/K, than the overall slope, ~ 0.1 meV/K

We verify this assignment by simulating the differential transmission in terms of the 3rd order nonlinear optical response: two interactions with the THz field and upconversion of the resulting microscopic second order polarization by the optical field (for further details see the Methods section)[39, 40]. Since we observe three distinct components in the time traces, we describe the data by a linear combination of three 3rd order processes–the electro-absorption[36], the phonon population and its relaxation (a monotonically decaying component), and the overtone phonon coherence and its dephasing (an oscillating component). For modeling the THz–phonon interaction and its temporal evolution, we use reported values for the center frequency and the bandwidth of the fundamental 1 THz phonon mode[20]. We further assume that the frequency difference between the $|0\rangle \rightarrow |1\rangle$ transition and $|1\rangle \rightarrow |2\rangle$ transition is negligible (harmonic approximation). To account for the dependence of electronic transitions on the phonon excitation, we describe the optical transitions by a Lorentzian band centered at ~ 1.63 eV with a bandwidth (FWHM, ~ 50 meV), obtained from the differential spectra at the long delay times ($t > 2$ ps, Fig. 1b). Despite the simplicity of this model, it excellently reproduces the relative phase of the three contributions for all probe frequency components in the measured data (see representative traces at 1.61 and 1.62 eV in Fig. 3). By extending this numerical analysis, we can rule out the possibility that another reported phonon mode, with the frequency of ~ 1.9 THz[18, 20–22], can produce the same transient differential spectrum (*blue dashed curves* in Fig. 3 and the details in the Methods section). So, our results show that the observed shift and oscillation of the band gap stem from the population and the overtone coherence of only the 1 THz mode, corresponding to the Pb-I-Pb angle bending phonon.

Having clarified the mechanism of each contribution to our time-dependent differential spectra, we examine the dynamic properties of this 1 THz phonon. The population relaxation time

of the 1 THz phonon, $\tau_{11}$ is determined to be $1.7 \pm 0.4$ ps at 295 K (see the Methods section), constituting the time scale of excess energy redistribution[38]. The dephasing time of the 1 THz phonon overtone, $\tau_{20}$, indicates the time scale of recovering the original state from the coherence between two states (i.e., states $|0\rangle$ and $|2\rangle$), and amounts to $0.78 \pm 0.14$ ps at 295 K (Supplementary Table 1). We obtained the value from the damping term of the oscillation, i.e., the width ($\Delta_{20}$) of the peak (dip) at $\sim 2$ THz in Fig. 1d ($\tau_{20} = 1/2\pi\Delta_{20}$). The overtone dephasing time of 1 THz phonon is comparable to the reported fundamental dephasing time $\tau_{10} \cong 0.67$ ps[20] within the experimental error. We note that both the fundamental and overtone dephasing times of 1 THz phonon are 3 ~ 5 times longer than the reported value of other energetically close Raman and infrared active phonon modes[20–22]. For example, the Raman active mode at 1.6 THz has a reported dephasing time of $\sim 0.1$ ps[21] and the 2 THz mode in THz absorption spectra of $\sim 0.25$ ps at room temperature[20].

**Temperature dependence of the coupling.** Next, we investigate this specific phonon-gap coupling at several other temperatures, since the properties of both the band gap and the phonon depend on temperature[20, 25]. For all studied temperatures (81, 185, 225, 265, 295 K), the monotonically decaying and oscillating contributions are observed as shown in Fig. 4a (and Supplementary Fig. 3). We note that the probe energies at which the transient differential spectra at $t > 1$ ps appear (exponentially decaying and oscillatory component) differ for different temperatures, which can be traced back to the shift of the optical gap with temperature (Figs. 4a and 5). The oscillatory components in the pump probe signals indicate that the responsible phonon mode for modulating the optical band gap remains the same as the one identified at 295 K—the Pb-I-Pb bending mode at 1 THz (Fig. 4b). The inferred central frequency and lifetime of the ~ 1 THz phonon exhibit a temperature dependence in good agreement with results from previous linear, steady-state spectroscopy (Fig. 4b, c)[20]. The center frequency of the 1 THz phonon mode is only slightly sensitive to temperature (varying within 0.94–0.95 THz in our experimental temperature range, 185–295 K) in the tetragonal phase[20]. For the measurement at 81 K, one has to realize that MAPbI3 undergoes a phase transition from the tetragonal phase (at room temperature) to the orthorhombic phase below ~ 160 K, accompanied by a band gap jump[25] and also by a splitting of the 1 THz phonon mode into two phonon modes at 0.75 and 0.95 THz[20]. While both phonons are resonant with the terahertz pulse, the phonon-induced modulation of the optical band gap occurs only at 1.5 THz. This reveals that the 0.75 THz mode is more strongly coupled to the optical gap than the 0.95 THz phonon (see Fig. 4b).

All transient spectra ($t > 1$ ps) at different temperatures in the tetragonal phase can be accurately represented by a similar blue shift of the linear absorption spectrum (Supplementary Fig. 4). Since we measured the temperature-dependent THz pump visible probe spectra inside the cryostat, the pump energy absorbed by the sample within the optical volume is much smaller than that in the free space (data in Fig. 1). Yet, the band gap shift ($\Delta E_{gap}$) is observed to be nearly insensitive to the experimental temperature in the tetragonal phase (185 ~ 300 K). At 81 K, the blue shift of the transmission curve explains the differential spectra only at the probe energies higher than ~ 1.68 eV, because the contribution of excitons to the transmission curve becomes non-negligible in this temperature[41]. Altogether with the temperature independence of the phonon temperature increase ($\Delta T_{ph}$), the temperature independence of the band gap shift ($\Delta E_{gap}$) indicates that band gap has a linear correlation with the 1 THz phonon population in the tetragonal phase (185 ~ 300 K).

## Discussion

Since the excitation of this 1 THz phonon consistently leads to an increase of the band gap at all experimental temperatures (in the tetragonal phase), we now connect this finding with the unusual temperature dependence of the band gap in the MAPbI3 perovskite. When the sample temperature increases, the population of all thermally accessible phonons increases and, in turn, the coordinates of these phonons are displaced. Nonetheless, the effect of each thermally populated phonon on the band gap depends on its coupling with the band gap. In case of the 1 THz phonon, we were able to selectively increase the phonon population and simultaneously observe the band gap shift: The THz excitation of our sample with the peak field strength of 100 kV/cm induces a blue-shift of the optical band gap by ~ 0.3 meV at 295 K (at $t = 0$, see Supplementary Fig. 1 and Methods section). The absorbed pump pulse energy is equivalent to the amount of heat that can increase the entire lattice temperature by $\Delta T_{avg} \sim 5 \times 10^{-3}$ K (Supplementary Eqn. 4), when both population of 1 THz phonon decays and the overtone dephases at longer times ($t > 4$ ps). The average temperature increase by energy dissipation ($\Delta T_{avg}$) has no noticeable contribution to the band gap shift as shown in Fig. 1b at $t > 4$ ps. This implies that other thermally accessible phonons are not strongly coupled to the band gap (compared to the 1 THz mode). When this energy is specifically absorbed by the 1 THz mode by resonant excitation, one can define an effective temperature increase of the 1 THz phonon $\Delta T_{ph}$. The phonon temperature $\Delta T_{ph}$, which identifies the population increase of only the 1 THz phonon, is estimated to be ~ 0.04 K (for the details see Methods section). We assume the phonon population of the 1 THz mode to obey a Boltzmann distribution after initial excitation (at $t = 0$). Based on this effective phonon temperature, the efficiency of the 1 THz phonon in increasing the band gap (with slope = $\Delta E_{gap}/\Delta T_{ph} \sim 7$ meV/K) is substantially larger than that of the steady-state thermal effect (slope = $\Delta E_{gap}/\Delta T_{actual} \sim 0.1$ meV/K in the measured sample, Fig. 5). We note that the potential negative contributions (i.e., the decrease of the band gap by increasing other phonon populations or lattice effects) are also expected, as commonly observed in semiconductors[23, 24]. Therefore, the comparison of these slopes implies that the thermal population of the 1 THz phonon has a considerable contribution to the temperature dependence of the band gap, compensating for other phonon mode population or lattice effects.

In conclusion, we have directly observed the mode-specific contribution of one phonon, the ~ 1 THz mode, to the position of the optical band gap in MAPbI3 perovskite. As this phonon mode has been assigned to the Pb-I-Pb angular bending vibration[21], our results provide direct experimental evidence for the theoretically predicted[18] correlation of the band gap with the Pb-I-Pb angle. Our finding that population of this 1 THz phonon increases the band gap is consistent with the unusual positive temperature dependence of the band gap[25]. The observed shift of the band gap caused by the number of absorbed photons implies a high efficiency of this phonon in increasing the band gap by thermal population. Therefore, our results demonstrate a mode-resolved approach to understanding the peculiar temperature dependence of the band gap in MAPbI3. Likewise, isolated phonon effects on photophysical properties of various semiconductors can be further explored through this approach without generating photocarriers, i.e., in the weak perturbation limit.

## Methods

**Sample preparation.** Polycrystalline MAPbI3 films were prepared by following standard procedures as described in previous studies[5, 42, 43]. Methylammonium iodide (MAI) was obtained by stirring methylamine (% 40 methanol) (27.86 mL) and hydrogen iodide (% 57 in water) (30 mL) at 0 °C for 2 h under nitrogen. The

solvent was removed by rotary evaporation at 50 °C, and the precipitated MAI was recrystallized from ethanol and diethylether. Crystalline MAI was recovered by suction filtration, and subsequently dried at 60 °C in a vacuum oven for 24 h[42]. We stored the thus obtained MAI in a nitrogen-filled glove box and a dark environment. MAPbI$_3$ perovskite films were prepared from a 40% (wt) solution of MAI and PbCl$_2$ in 3:1 molar ratio in anhydrous N,N-dimethylformamide and deposited on the fused silica substrates by spin coating at 2000 rpm in a nitrogen filled glove box. After spin-coating, the films were left to dry at room temperature for 30 min and subsequently annealed for 45 min on a 100 °C hotplate in the glove box[43].

Structural analysis of MAPbI$_3$ perovskite films was performed by X-ray diffraction (XRD, Bruker D8 Advance 2 / θθ -Diffractometer). XRD patterns of MAPbI$_3$ perovskite films (Supplementary Fig. 5) show (110), (211), (220), (310), and (330) reflections, which are characteristic for the pristine perovskite tetragonal crystal structure at room temperature[44, 45]. The absorption spectra of the films (Supplementary Fig. 6) reveal a band gap onset at $1.60 \pm 0.03$ eV that were measured using a ultraviolet–visible spectrometer (Perkin Elmer Lambda 900). The high background below the band gap can be attributed to light scattering by large micrometer-sized organometal halide perovskite particles[46]. The films had an average thickness of 300 nm (rms roughness of 75 nm) as measured by a Tencor P-10 Step Profiler using a 3500 µm scan length and a scan speed of 20 µm/s. Scanning electron microscopy images show micron sized polycrystals in MAPbI$_3$ perovskite films (Supplementary Fig. 7). We have shown in our previous study[43] that the lifetime of the charge carriers determined by time resolved photoluminescence ($129.1 \pm 0.7$ ns)[43] is within the—broad range of values—reported in literature (1–300 ns)[5, 7, 15, 47]. Using time-resolved terahertz spectroscopy measurements, we have shown that the photo-carrier mobility in our solution processed perovskite samples is ~ $27 \pm 3$ cm$^2$/V/s, which is consistent with reported THz mobilities (30–35 cm$^2$/V/s)[47, 48] for an upper threshold limit for solution processed MAPbI$_3$ crystals[43]. These similar mobilities verify the quality of our samples compared to other studies.

**THz pump visible probe spectroscopy.** Sub-ps single-cycle THz pulses with a peak field strength of ~ 100 kV/cm were generated using a tilted wave-front scheme for optical rectification[49, 50]. Using a grating with 2000 /mm we tilt the wave-front of 800 nm pulses with a duration of 130 fs and a pulse energy of 3.2 mJ (1 kHz repetition rate). These pulses were transmitted through a LiNbO$_3$ crystal such that the generated THz pulses propagate at normal incidence from the surface. The THz field amplitude was controlled by a pair of wire-grid polarizers and focused on the sample by a pair of parabolic mirrors with a focal length of three inches. The resulting THz field was characterized by electro-optic sampling in a ZnTe crystal using balanced detection of the gating pulses[51]. For detection of the THz-induced optical absorption changes, we generate white light supercontinuum pulses by focusing 800 nm pulses into a sapphire crystal. The continuum probe pulse were focused roughly at the sample position to achieve an optimal spatial-overlap with the THz pump-pulse and re-collimated after the sample. We spectrally resolved the probe pulses at wavelengths ranging from 450 to 780 nm by dispersing them onto a metal oxide semiconductor (NMOS) camera using an imaging spectrograph. A fraction of the visible probe pulse was split off before the sample and dispersed on a second NMOS array to correct for pulse-to-pulse fluctuations. The relative timing between the THz pump and the visible probe pulse was controlled using a translational stage. To detect the differential optical transmission spectra through the sample we use single pulse detection and modulate the pump pulse with an optical chopper. For measurements at 295 K, the THz beam path and the sample compartment in the spectrometer were purged with dry nitrogen in order to avoid sample degradation. Temperature-dependent measurements were performed in a vacuum chamber of a liquid nitrogen cooled cryostat.

**Numerical analysis.** We simulated the differential transmission dynamics in terms of the 3rd order nonlinear polarization (Eq. 1)[40] to confirm the qualitative picture of the 1 THz phonon-mediated optical band gap modulation.

$$\frac{\Delta}{T} \propto iP^{(3)}(t) = i \int_0^\infty dt_3 \int_0^\infty dt_2 \int_0^\infty dt_1 R^{(3)}(t_3, t_2, t_1) E_{vis}(t-t_3) E_{THz}(t-t_3-t_2) E_{THz}(t-t_3-t_2-t_1) \tag{1}$$

where $t_n$ indicates time intervals between the field-matter interactions. $t_1$ is the earliest time interval (i.e., time interval between the first and the second THz-perovskite interaction).

Besides the Pb-I-Pb angle bending mode (~ 1 THz), MAPbI$_3$ perovskites exhibit another phonon mode at ~ 1.9 THz with significant Raman activity[18, 21, 22]. In case of (i) the phonon at 1 THz mode modulating the optical transmission, the resultant 2 THz coherence is the coherent superposition of the vibrational ground ($v = 0$) and the second excited ($v = 2$) state. The coherence is attained by the second interaction between the field and the fundamental coherence after the first THz-perovskite interaction. Alternatively, (ii) the 1.9 THz mode can be excited via virtual states, the superposition of $v=0$ and 1 of the 1.9 THz mode is produced via the polarizability-field interactions, as commonly seen in the traditional coherent phonon spectroscopic approaches. Thus, the response function, $R^{(3)}$, for each pathway was modeled by the equations of motion for the density matrix in a 4-level system with a general relaxation mechanism as described by the following set of equations. For the overtone coherence generation ((i), 1 THz mode) pathway,

$$R^{(3)}(t_3, t_2, t_1) \propto \left(\frac{i}{\hbar}\right)^3 e^{i\omega_{30}t_3 - \Gamma_{30}t_3} e^{i\omega_{20}t_2 - \Gamma_{20}t_2} e^{i\omega_{10}t_1 - \Gamma_{10}t_1} \tag{2}$$

where $\omega_{k0}$ is angular frequency, corresponding to the energy difference between the state k and 0, and $\Gamma_{k0}$ is dephasing rate for the superposition of the state k and 0. We use the reported value, $\Gamma_{10} = 1.5$ THz[20], for the phonon scattering rate, the center frequency obtained from the damped oscillator fit, $\omega_{20} = 1.94$ THz, for the overtone frequency of 1 THz mode, and assume a harmonic oscillator ($\omega_{20} = 2 \times \omega_{10}$).

To model the population relaxation, we used

$$R^{(3)}(t_3, t_2, t_1) \propto \left(\frac{i}{\hbar}\right)^3 e^{i\omega_{31}t_3 - \Gamma_{31}t_3} e^{-\Gamma_{11}t_2} e^{i\omega_{10}t_1 - \Gamma_{10}t_1} \tag{3}$$

where the population decay time, $1/\Gamma_{11} = 1.7$ ps, was obtained from the damped oscillator fit.

For the non-resonant ((ii), 1.9 THz mode) pathway, we used

$$R^{(3)}(t_3, t_2, t_1) \propto \left(\frac{i}{\hbar}\right)^3 e^{i\omega_{30}t_3 - \Gamma_{30}t_3} e^{i\omega_{10}t_2 - \Gamma_{10}t_2} e^{i\omega_{i0}t_1 - \Gamma_{i0}t_1} \tag{4}$$

where $\omega_{k0}$ is the angular frequency, corresponding to the energy difference between the state k and 0, $\Gamma_{k0}$ is dephasing rate for the superposition of the state k and 0 in case of the 1.9 THz phonon mode, and $i$ indicates a virtual state which has a dephasing rate >> 1.5 THz and a center frequency >> 1 THz.

For both scenarios (i and ii), we used $\Gamma_{30} \cong \Gamma_{31} = $ ~50 meV and $\omega_{30} \cong \omega_{31} = $ ~1.63 eV to model the optical transitions, which was obtained from the differential spectra at long delay times ($t > 2$ ps). The instantaneous contribution, which is proportional to the square of the THz transient, was added as an additional 3rd order process.

For extracting phenomenological information from our pump probe signal, we adopted a simple damped oscillator model.

$$-\Delta T/T = A_{exp} e^{-\Gamma_{11}t} + A_{osc} e^{i\omega_{20}t - \Gamma_{20}t + \varphi} \tag{5}$$

We decomposed the two components of the pump probe signal—an exponentially decaying population and a dephasing wave packet oscillation—present at long delay times ($t > 1.3$ ps) by fitting this model to our data (at $t > 1.3$ ps). The thus obtained exponential relaxation time, $1/\Gamma_{11}$, the oscillation frequency, $\omega_{20}$, and the damping time, $1/\Gamma_{20}$ for each experimental condition is given in Table S1. The values reported in Table S1 represent weighted averages over detection frequencies ranging from 1.6 to 1.67 eV (at 295 K) using the amplitudes of the contribution as weights ($w_i = A_{exp}$ or $A_{osc}$).

$$\overline{\beta} = \frac{\sum_i w_i \cdot \beta_i}{\sum_i w_i} \tag{6}$$

where $\beta_i (= \Gamma_{11}, \Gamma_{20},$ or $\omega_{20})$ is the obtained parameter at a given detection energy and $\overline{\beta}$ is average value (Table S1). The error ranges, $\sigma$, listed in Table S1 corresponds to the weighted standard deviation of the determined parameter at different detection energies:

$$\sigma = \frac{\sqrt{\left(\sum_i w_i \cdot \left(\beta_i - \overline{\beta}\right)^2\right)}}{\sum_i w_i} \tag{7}$$

**Estimation of THz-induced temperature rise.** To estimate the effective temperature increase of the 1 THz phonon (temperature equivalence of the population increment for the 1 THz phonon, $\Delta T_{ph}$), we assume that the phonon population follows the Boltzmann distribution both before and after the THz excitation as follows:

The probability for population of the $n$th phonon level is

$$P_n = \frac{1}{Z} \exp\left(-\frac{nh\nu}{k_B T}\right) \tag{8}$$

where $Z = \frac{1}{1 - \exp\left(-\frac{h\nu}{k_B T}\right)}$.

At the initial temperature $T_i$ (before THz pulse), the total energy of this phonon system is

$$U_i = \sum_{n=0}^\infty nh\nu \cdot P_{n,i} \cdot N_{tot} \tag{9}$$

At the resultant temperature $T_f$ after absorption of the THz (additional $\Delta E_{Abs}$), the total energy of this phonon system is

$$U_f = \sum_{n=0}^\infty nh\nu \cdot P_{n,f} \cdot N_{tot} = U_i + \Delta E_{Abs} \tag{10}$$

Therefore,

$$T_{\mathrm{f}} = \frac{h\nu}{k_{\mathrm{B}}}\left(\log\left(\frac{1}{\frac{1}{\exp\left(\frac{h\nu}{k_{\mathrm{B}}T_{\mathrm{i}}}\right)-1}+\frac{\Delta E_{\mathrm{Abs}}}{h\nu N_{\mathrm{tot}}}}+1\right)\right)^{-1} \quad (11)$$

Based on the basic parameters from our experimental condition (Supplementary Note 1), we have the total number of oscillators in the optical volume $N_{\mathrm{tot}} = 2.7894 \times 10^{15}$, the absorbed THz energy $\Delta E_{\mathrm{Abs}} = 1.4923 \times 10^{-9}$[J], and the exact value of the center frequency of 1 THz phonon $\nu = 0.94 \times 10^{12}$[Hz]. The exact value of the center frequency of the 1 THz phonon, 0.94 THz, was obtained by THz absorption measurement on our sample (Supplementary Fig. 8). The center frequency of this 1 THz phonon mode is nearly insensitive to temperature (varying within 0.94–0.95 THz in our experimental temperature range, 185–295 K) in the tetragonal phase[20]. Thus, given the same amount of absorbed THz energy, the phonon temperature increase ($\Delta T_{\mathrm{ph}}$) is considered to be independent on the experimental temperature (Eq. 11 and Supplementary Fig. 9). Therefore,

$$\Delta T_{\mathrm{ph}} = T_{\mathrm{f}} - T_{\mathrm{i}} \sim 0.04\,\mathrm{K} \quad (12)$$

**Estimation of the efficiency of the 1 THz phonon in increasing the band gap**. We compare the corresponding temperature rise to the transient shift of the optical band gap due to the THz excitation. For the direct correlation between the temperature and the optical band gap, we extracted the temperature dependence of the band gap using a linear fit to the extracted data at 185 to 300 K (in the tetragonal phase) from our THz pump-optical probe experiments:

$$E_{\mathrm{gap}}[\mathrm{eV}] = 1.33 \times 10^{-4} \cdot T_{\mathrm{actual}}[\mathrm{K}] + 1.574 \quad (13)$$

From our transient signals, we estimate the optical band gap shift by shifting the measured visible transmission spectrum and comparing the difference between the measured (Supplementary Fig. 1a, *black*) and the shifted (Supplementary Fig. 1a, *green*) spectra to the transient signal. As can be seen from Supplementary Fig. 1b, best agreement between the experimental transient spectra and the estimated shift of the bandgap is obtained using a shift of the optical band gap by $\Delta E_{\mathrm{gap}} \sim 0.3$ meV. We note that, in order to account for the decay of the transient signals, we multiply the experimental transient spectra with a factor of $1/e^{(t/\tau_{11})}$. Also, we selected the delay times where the oscillatory component has negligible contribution. Altogether with the temperature dependence of the optical band gap[25], we thus estimated the efficiency of the 1 THz phonon in increasing the band gap of

$$\Delta E_{\mathrm{gap}}/\Delta T_{\mathrm{ph}} \sim 7\,\mathrm{meV/K} \quad (14)$$

as measured by the shift of the optical band gap.

**Data availability**. The data that support the findings of this study are available from the corresponding author upon reasonable request.

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

## Acknowledgements

H.K. acknowledges financial support from the European Union's Horizon 2020 research and innovation program under grant agreement No.658467. D.T. acknowledges the EU Career Integration Grant (CIG 334324 LIGHTER). We thank Marc-Jan van Zadel for technical support and Yuki Nagata for valuable discussions.

## Author contributions

M.B., J.H., H.K., and E.C. conceived the project and designed the experiments. Z.M., D.T., S.H.P., J.H., and H.K. constructed the experimental set-up and H.K. performed the experiments. M.K. and E.C. prepared and characterized the perovskite samples. H.K., M. B., J.H., S.H.P., and M.G. interpreted the results. H.K., M.B., and J.H. analyzed the experimental results and M.G., H.K., J.H., and M.B. developed the models. H.K., J.H., E.C., S.H.P., M.G., D.T., and M.B. wrote the paper.

## Additional information

**Competing interests:** The authors declare no competing financial interests.

