## [Peer Review File · Nature Communications]

Reviewers' comments:

Reviewer #1 (Remarks to the Author):

The authors probe the dependence of the interband absorption of methylammonium lead iodide on excitation by strong THz pulses. They find that the band gap is strongly influenced by excitation of a phonon near 1 THz, which is associated with collective motions of Pb-I-Pb ions. The authors argue that these observations explain the surprising fact that the band gap of this material increases with increasing temperature. The data are convincing, the manuscript is well-written, and the conclusions are important and well-supported. I recommend publication as-is, my only suggestion is that the authors add a scale bar to Fig. 1c.

Reviewer #2 (Remarks to the Author):

In this manuscript the authors investigate the origin of the anomalous temperature dependence of the band gap of MAPbI₃. Using transient absorption measurements they propose that the band gap couples strongly to a lattice vibrational mode at 1 THz, which would correspond to the bending motion of Pb-I-Pb bonds in the inorganic sublattice of this perovskite.

This topic is of immediate interest to the community working on hybrid perovskites and related materials, and the anomalous band gap shift of MAPbI₃ is an intriguing effect which currently lacks a solid understanding.

The approach pursued by the authors is interesting, however in my view they overinterpret their data and draw conclusions that are not supported by the evidence at hand. I elaborate on these aspects below.

The main claim of this manuscript, that the temperature dependence of the band gap of MAPbI₃ results from a strong electron-vibration coupling to a single phonon at 1 THz, is mostly speculative and does not stand up to close scrutiny.

These shortcomings are serious enough that I cannot recommend the manuscript for publication in Nature Communications.

1) Throughout the manuscript the authors refer to "one phonon" at 1 THz which (quoting from the manuscript) "fully and uniquely" accounts for the temperature dependence of the band gap of MAPbI₃.

This very strong claim is not supported by their data: what the authors are proving is that after optically pumping *some* phonons in the neighborhood of 1 THz they obtain a small blueshift of the optical absorption, corresponding to *approximately* 0.3 meV.

Now the temperature variation of the gap of MAPbI₃ across the tetragonal

phase is as large as 40 meV, therefore their inference is decidedly far fetched.

In order to obtain the complete temperature dependence of the band gap of MAPbI₃ from the 0.3 meV shift they state that following the laser pulse the temperature of the sample increases by *approximately* 1 K. From the values 0.3 meV and 1 K, by *assuming* that only one mode is thermally excited, they deduce the population of this mode, and use the rate thus obtained of *approximately* 0.3 meV / 1 K to explain a variation of the band gap of 40 meV over 150 K.

Apart from the unjustified assumption regarding the selective population of a single phonon, even by only considering some error bars on their data one would see that they might miss the observed temperature dependence by a large factor, simply because they are extrapolating from very small numbers to large numbers.

2) In the manuscript the authors do not report the temperature dependence of the band gap, and use instead previously reported data. Now the temperature shift of the gap of MAPbI₃ is sample-dependent, for example [AFM 25 6218 (2015)] and [10.1126/sciadv.1601156] report shifts of 30 meV and 40 meV across the tetragonal phase. In addition the temperature dependence in the orthorhombic phase is rather more complex than a straight line [Nat. Commun. 7, 11755 (2016)], yet the authors propose that their interpretation is also valid at low temperature.

3) In the conclusions the authors claim that the broadening of the absorption edge is accounted for by the interaction between electrons and one phonon mode at 1 THz. This is in strong disagreement with the fact that the broadening of the absorption lineshapes was shown to arise from a Froehlich interaction with LO phonons at 11 meV [Nat. Commun. 7, 11755 (2016)]. The 1 THz phonon considered by the authors in this work is incompatible with the temperature dependence of the broadening reported in the cited work, which rules out their proposed mechanism.

4) In [10.1126/sciadv.1601156] the temperature dependence of the gap was shown to arise from lattice thermal expansion. The authors do not discuss the disagreement with this previous assignment, and do not provide any evidence to show how their proposed scenario is to be preferred to the one of the cited work.

5) The discussion lacks any reference to excitons. The authors are probing optical transitions right at the edge of the first bound exciton, but no discussion is provided on the effect of electron-hole interactions on their measurements, interpretation, and conclusions.

6) The Franck-Condon diagram sketched in Fig. 3 is not relevant and misleading. The electronic structure of the inorganic sublattice of MAPbI₃ cannot be compared to a molecular system, in fact it is much closer to a III-V semiconductor like GaAs. In particular the potential energy surfaces in the ground state and in the excited state of MAPbI₃ differ only by a rigid shift and do not have distinct minima. In fact the exciton binding energies are very small (10-30 meV depending on the measurement) hence this system resides in the Wannier regime, not the Frenkel regime as Fig. 3 would imply.

Reviewer #3 (Remarks to the Author):

The manuscript describe a bandgap bleaching features with oscillation of ~ 1 THz by ultrafast near-resonant THz phonon excitation in a perovskite semiconductor. With the help of simulations, authors explains the temperature dependence of the gap in the tetragonal phase as the thermal population of the 1 THz phonon mode is the major cause of thermal-shift in the bandgap. The experiment is well designed and results are intriguing. However, there are a few issues and concerns below that should be adequately addressed.

(1) The coherent phonons oscillation seen in the white light spectra is very convincing to conclude that there is a strong coupling of the lattice and electronic degrees of freedom. However, it is of much more speculative to further extend this to the coupling between a specific phonon and optical properties. The authors provide arguments mostly based on indirect evidence such as estimations of temperatures increase and subtle phase difference. However, such a claim really calls a clear cut experiment feature, e.g., if authors can shift excitation spectra to 2THz Does it give the same effect to the band gap? It is probably technically difficult to do that. Anyway a more convincing data is needed to support the mode-specific nature. There are more problems in the data for such a claim, e.g., temperature dependence of modulation frequency changes from 2THz to ~ 1.8 THz, does this correlate with the temperatures dependence shift of phonons? I suggest authors improve this and further justify the claim which can be a substantial progress.

(2) The modulation is highly damped in the time domain (< 4 ps) which produce a broaden width in the FFT spectra at least twice as large as compared to the linear THz absorption (~ 1 meV). This should be explained more.

(3) In several places of the text the authors seems to imply the ultrafast band gap shift as thermal effect "that the excess energy absorbed by specific excitation of the lattice phonon." This is confusing to me. It seems the blue absorption decay correlates with the dephasing of the phonons, i.e., coherence time. This is inconsistent with the energy dissipation or thermal picture of the 1THz phonon. A related question: If the lattice is oscillating back and forth (or fully coherent), I would expect the near-band-gap absorption oscillate near zero. Why is there a finite blue shift in the band edge in the entire temporal profile?

(4) Figure s1: why is the absorbance still very high below the band gap of ~ 1.6 eV?

(5) Fig S6 and S7 – these two data sets are very interesting and relevant for the conclusion if explained well. I don't understand why authors don't put them in the main text. What is the origin of the phonon splitting at low temperatures? Are these two modes from the same 1THz phonon or they are separate from different origins? Are there any theory to calculate on their effects on the band structure?

In summary, this is a potentially interesting paper and target relevant questions for the community. However, there are some substantial issues, as listed above, which should be clarified.

Reviewers' comments:

Reviewer #2 (Remarks to the Author):

I commend the authors for addressing all the comments raised by myself and the other referees.

I consider that the manuscript has improved substantially, however the authors are still exaggerating some of their claims:

In their rebuttal they agree with all my comments, and even admit that their previous estimate of the electron-phonon coupling strength was rather inaccurate. The revised coupling strength that they now find is 15 times larger than what is needed to explain the temperature dependence of the band gap. By using this value one obtains a band gap variation of 0.6 eV across the tetragonal phase of MAPI, which is obviously in strong disagreement with every experiment reported so far.

The authors address this point in their rebuttal, but they seem unwilling to accordingly tone down their claims in the manuscript. Quoting from the abstract and from the conclusions:

"Excitation of the 1 THz phonon causes a blue shift of the band gap, explaining the temperature dependence of the gap in the tetragonal phase."

"Hence, our results provide a rationale for the peculiar temperature dependence of the band gap in MAPbI₃."

These statements are in contradiction with the author's rebuttal, and are not supported by this study.

There is no doubt that this is an interesting work, but I strongly recommend that the authors revisit their statements in the manuscript in order to offer a fairer interpretation and to avoid unnecessary and misleading claims.

Reviewer #3 (Remarks to the Author):

The authors have answered most of my questions satisfactorily. The revised manuscript is also significantly improved. There are two remaining issues regarding my prior comments: (1) selective pumping to match the phonon spectra. The authors answered this with figure S4. This is confusing. If one detune the pump away from 1 THz phonon, the effect should be diminished. Did author try this, e.g., filter the pulse to 0.5 THz or 1.5 THz? (2) If the population of the 1 THz mode causes the band shift, then one will expect a rise time instead of decay in Fig. 2b, since the population gets larger when the coherence is

dephased, which in turn case bigger band shift. It will be good to clarify this.
After these remaining issues clarified, this work should be published.

REVIEWERS' COMMENTS:

Reviewer #3 (Remarks to the Author):

I found the authors' answer to my second question of phonon population vs band shift is quite speculative and confusing. I don't understand how one can talk about the temperature if the decayed state has no population change in the reservoir. This is important to be clarified since it also relates to the other referee's question on the exact nature of band shift vs phonon temperature. I have no problem for the quality of the data but the paper can be benefited for more clarifications on these issues.

Reviewers' comments:

Reviewer #1 (Remarks to the Author):

The authors probe the dependence of the interband absorption of methylammonium lead iodide on excitation by strong THz pulses. They find that the band gap is strongly influenced by excitation of a phonon near 1 THz, which is associated with collective motions of Pb-I-Pb ions. The authors argue that these observations explain the surprising fact that the band gap of this material increases with increasing temperature. The data are convincing, the manuscript is well-written, and the conclusions are important and well-supported. I recommend publication as-is, my only suggestion is that the authors add a scale bar to Fig. 1c.

→ We thank the reviewer very much for the encouraging comments and suggestion. We have added a scale bar to Figure 1c.

Reviewer #2 (Remarks to the Author):

In this manuscript the authors investigate the origin of the anomalous temperature dependence of the band gap of MAPbI₃. Using transient absorption measurements they propose that the band gap couples strongly to a lattice vibrational mode at 1 THz, which would correspond to the bending motion of Pb-I-Pb bonds in the inorganic sublattice of this perovskite.

This topic is of immediate interest to the community working on hybrid perovskites and related materials, and the anomalous band gap shift of MAPbI₃ is an intriguing effect which currently lacks a solid understanding.

The approach pursued by the authors is interesting, however in my view they overinterpret their data and draw conclusions that are not supported by the evidence at hand. I elaborate on these aspects below. The main claim of this manuscript, that the temperature dependence of the band gap of MAPbI₃ results from a strong electron-vibration coupling to a single phonon at 1 THz, is mostly speculative and does not stand up to close scrutiny.

These shortcomings are serious enough that I cannot recommend the manuscript for publication in Nature Communications.

1) Throughout the manuscript the authors refer to "one phonon" at 1 THz which (quoting from the manuscript) "fully and uniquely" accounts for the temperature dependence of the band gap of MAPbI₃. This very strong claim is not supported by their data: what the authors are proving is that after optically pumping *some* phonons in the neighborhood of 1 THz they obtain a small blueshift of the optical absorption, corresponding to *approximately* 0.3 meV.

Now the temperature variation of the gap of MAPbI₃ across the tetragonal phase is as large as 40 meV, therefore their inference is decidedly far fetched.

In order to obtain the complete temperature dependence of the band gap of MAPbI₃ from the 0.3 meV shift they state that following the laser pulse the temperature of the sample increases by *approximately* 1 K. From the values 0.3 meV and 1 K, by *assuming* that only one mode is thermally excited, they deduce the population of this mode, and use the rate thus obtained of *approximately* 0.3 meV / 1 K to explain a variation of the band gap of 40 meV over 150 K.

Apart from the unjustified assumption regarding the selective population of a single phonon, even by only considering some error bars on their data one would see that they might miss the observed temperature dependence by a large factor, simply because they are extrapolating from very small numbers to large numbers.

→ We very much appreciate the reviewer's concerns and detailed questions. While some of the issues raised can be addressed by improved presentation on our part, we agree with the reviewer that we should have questioned our interpretation more critically. However, our main finding in this work is the direct observation of a mode-specific effect of one phonon on the band gap: the population of this single phonon increases the band gap and this behavior is consistent with the temperature dependent behavior of the band gap. This qualitative behavior itself, we submit, is in itself remarkable, but we agree that we should have been more careful about the quantitative interpretation.

The reviewer raises a few specific points:

some phonons in the neighborhood of 1 THz

We note that while the excitation takes place via a broadband terahertz pulse, so that all optical phonons within the bandwidth (0.2 ~ 1.8 THz) can be excited, the bandgap modulation only oscillates at precisely one frequency. Accordingly, this shows that a single phonon mode *must* be responsible; any combination of modes will produce beats, which are not observed in our experiments. Therefore the responsible phonon frequency is well-defined. The uncertainty on the inferred frequency ~ 10 GHz according to the damped oscillator fit (see SI, Table S1).

corresponding to *approximately* 0.3 meV.

This is also well-defined number, which can be obtained directly from the amplitude of the differential spectra, to within a few percent.

Now the temperature variation of the gap of MAPbI₃ across the tetragonal phase is as large as 40 meV, therefore their inference is decidedly far fetched.

We apologize for our initial unclear presentation of this information. The essential point is not the absolute magnitude of the 0.3meV shift relative to the 40meV shift, but rather the fact that the very low excitation densities, which correspond to an increase in phonon temperature of at most 1K (see below), induce the 0.3meV shift. The relevant quantity for comparison is therefore 0.3meV/1K, which is similar to or even higher than the steady-state shift in tetragonal phase of 40meV/~150K.

In the revised manuscript, we include new data showing that the band gap shift (ΔE_{gap}) induced by the same amount of absorbed pump energy is nearly independent on the experimental temperature (Figure

S9 and the corresponding paragraph in SI); and new analysis showing that the phonon temperature increment (ΔT_{ph}) are not sensitive to the experimental temperature of the sample (185, 225, 265, 300 K). The independence of both ΔE_{gap} and ΔT_{ph} on temperature indicates that the slope of $\Delta E_{gap}/\Delta T_{ph}$ is nearly constant over the entire temperature range in the tetragonal phase (linear correlation between ΔE_{gap} and ΔT_{ph}). We have included these analyses in SI (Figure S6 and S9). We have tried to further clarify this key point in our revised manuscript.

temperature of the sample increases by *approximately* 1 K

As noted above, we were apparently not sufficiently clear. The temperature of the entire sample does not increase by this ~ 1 K; rather, the effective temperature of only the 1 THz phonon mode (the temperature equivalence of the population increment for the 1 THz mode).

Apart from the unjustified assumption regarding the selective population of a single phonon, even by only considering some error bars on their data one would see that they might miss the observed temperature dependence by a large factor, simply because they are extrapolating from very small numbers to large numbers.

We thank the reviewer for the concern. We agree that the slopes of $\Delta E_{gap}/\Delta T_{ph}$ at different temperatures should also be compared with that at 295 K, so that we have further support. As we explained above and the SI, we observe the independence of both ΔE_{gap} and ΔT_{ph} on temperature (which shows the expected linear correlation between ΔE_{gap} and ΔT_{ph}) over the entire temperature range in the tetragonal phase.

The reviewer's overall comments did, however, prompt us to try to be more accurate about the temperature increment in the phonon mode. We have improved our model, taking all higher quantum levels of this 1 THz phonon into account for the Boltzmann distribution, so that we can obtain a more reliable estimate of the effective temperature of this 1 THz mode after the THz absorption (Section 4-2 in the supplementary information). Using this model, we are also able to obtain the phonon temperature increments (ΔT_{ph}) for different initial experimental temperatures (Figure S6). These calculations reveal that ΔT_{ph} is not sensitive to the initial temperature in the experiment (81-300 K, Figure S6), with the same amount of absorbed THz energy. The more reliable estimate of the phonon temperature increment (ΔT_{ph}) is ~ 0.2 K, which yields the slope of $\Delta E_{gap}/\Delta T_{ph} \sim 1.5 \text{ meV K}^{-1}$ together with the observed $\Delta E_{gap} \sim 0.3 \text{ meV}$. We also note, in the revised manuscript, that, while one can define an effective phonon temperature, the terahertz pulse in the experiment intrinsically generates a non-equilibrium state, which can only be approximated by temperature increase ΔT_{ph} . In any case, the inferred $\Delta E_{gap}/\Delta T_{ph} \sim 1.5 \text{ meV K}^{-1}$ is larger than the steady-state slope of $\Delta E_{gap}/\Delta T_{sample} \sim 0.1 \text{ meV K}^{-1}$, which we have obtained specifically for our sample (Figure 5). We discuss this estimation in detail in the SI and we have revised the manuscript as follows:

“Since the increased population of this 1 THz phonon consistently leads to the increase of the band gap at all experimental temperatures (in the tetragonal phase), we now connect this finding with the unusual temperature dependence of the band gap in the MAPbI3 perovskite. When the sample temperature

increases, the population of all thermally accessible phonons increases and, in turn, the coordinates of these phonons are displaced. However, each phonon has different potential contribution to the band gap shift by its population depending on its coupling with the band gap. In case of this 1 THz phonon, we were able to selectively increase the phonon population and simultaneously observe the band gap shift: The THz excitation with the peak field strength of 100 kV/cm on our sample induces a blue-shift of the optical band gap by ~ 0.3 meV at 295 K (at $t = 0$, see SI). Although the selective excitation of the 1 THz phonon inherently leads to a non-equilibrium situation, one can define an effective temperature increase of the 1 THz phonon ΔT_{ph} , which can be estimated to be ~ 0.2 K (for the details see SI). Within the assumption of the Boltzmann distribution within the 1 THz mode, the efficiency of the 1 THz phonon in increasing the band gap (the slope = $(\Delta E_{gap})/(\Delta T_{ph}) \sim 1.5$ meV K⁻¹) is larger than that of the steady-state thermal effect (the slope = $(\Delta E_{gap})/(\Delta T_{actual}) \sim 0.1$ meV K⁻¹) in our sample, Figure 5). This qualitative comparison of these slopes indicates that the thermal population of the 1 THz phonon is expected to have a significant contribution to the temperature dependence of the band gap, compensating potentially for other phonon mode population or lattice effects."

In summary, we appreciate the reviewer's concern that our initial conclusion/claim is too strong, so we have removed potentially overstated expressions such as "one phonon at 1 THz "fully and uniquely" accounts for the temperature dependence of the band gap of MAPbI3." Accordingly, we have modified our conclusion as follows:

"In conclusion, we have directly observed the mode-specific contribution of one phonon, the ~ 1 THz mode, to position of the optical band gap in MAPbI3 perovskite.

...

Our finding that population of this 1 THz phonon increases the band gap is consistent with the positive temperature dependence of the band gap. Also, the observed shift of the band gap by the given number of absorbed photons indicates high efficiency of this phonon in increasing the band gap by thermal population.

...."

2) In the manuscript the authors do not report the temperature dependence of the band gap, and use instead previously reported data. Now the temperature shift of the gap of MAPbI3 is sample-dependent, for example [AFM 25 6218 (2015)] and [10.1126/sciadv.1601156] report shifts of 30 meV and 40 meV across the tetragonal phase. In addition the temperature dependence in the orthorhombic phase is rather more complex than a straight line [Nat. Commun. 7, 11755 (2016)], yet the authors propose that their interpretation is also valid at low temperature.

→ Thank you for the comment, which is also very relevant for the quantitative comparison. In the revised manuscript, rather than using previously reported data, we now use the temperature dependent measurements of the bandgap recorded for the same samples used in the THz pump-optical probe experiments. From these data, we have obtained the slope and we have replaced the reported

(reference) with the data from our sample (Figure 5). Our sample indeed exhibits a slightly lower slope than the reported one.

Please note, however, that the phonon-gap coupling itself has been also observed in 81 K (Figure 4), and this is what we mean when stating “valid at low temperature”. We now better describe the temperature dependent TPOP in the following paragraph:

“Next, we investigate this specific phonon-gap coupling at several other temperatures, since the properties of both the band gap and the phonon depend on temperature. For all studied temperatures (81, 185, 225, 265, 295 K), the monotonically decaying and oscillating contributions are observed as shown in Figure 4a (and Figure S8). For the highest three temperatures, the terahertz-induced bandgap shift is very similar to that at 185K (see Figure S9), with the inferred central frequency and lifetime of the ~1 THz phonon exhibiting a temperature dependence in good agreement with results from previous linear, steady-state spectroscopy (Figures 4b and 4c). All transient spectra ($t > 1$ ps) at different temperatures in the tetragonal phase can be accurately represented by a similar blue shift of the linear absorption spectrum (Figure S9). For the measurement at 81 K, one has to realize that MAPbI₃ undergoes a phase transition from the tetragonal phase (at room temperature) to the orthorhombic phase below ~160 K, which is accompanied by a band gap jump and also by a splitting of the 1 THz phonon mode into two phonon modes at 0.75 and 0.95 THz. While both phonons are excited by the terahertz pulse, the phonon-induced modulation of the optical band gap occurs only at 1.5 THz, which reveals that only the 0.75 THz (and not the 0.95 THz phonon) mode is strongly coupled to the optical gap (see Figure 4b).”

We have revised the main text by clarifying exact range of temperature instead of using “low-temperature”.

3) In the conclusions the authors claim that the broadening of the absorption edge is accounted for by the interaction between electrons and one phonon mode at 1 THz. This is in strong disagreement with the fact that the broadening of the absorption lineshapes was shown to arise from a Froehlich interaction with LO phonons at 11 meV [Nat. Commun. 7, 11755 (2016)]. The 1 THz phonon considered by the authors in this work is incompatible with the temperature dependence of the broadening reported in the cited work, which rules out their proposed mechanism.

→ Thank you for this comment. You are absolutely right. We do not have a clear evidence of the exact anharmonicity of the 1 THz phonon mode and the actual broadening of the band gap so that we cannot explicitly/quantitatively discuss about the broadening of the absorption line-shape contributed by the 1 THz mode. Therefore, we removed all the discussion about the broadening. We cite the mentioned work for the following statement in the introduction.

“Additionally, thermal population of phonons in MAPbI₃ perovskites is expected to dominate the electronic disorder, as the contribution of lattice displacements outweighs defect or impurity-related contributions to the lineshape of photoluminescence.”

4) In [10.1126/sciadv.1601156] the temperature dependence of the gap was shown to arise from lattice thermal expansion. The authors do not discuss the disagreement with this previous assignment, and do not provide any evidence to show how their proposed scenario is to be preferred to the one of the cited work.

→ Thank you very much for introducing this work. This work is clearly relevant to our experimental finding and indeed interesting. However, we can assure that this work is not in conflict with our finding, even though they indeed observed the band gap increase as a function of a pseudocubic lattice parameter, $a = \sqrt[3]{V}$ (Figure 5b in [10.1126/sciadv.1601156]). The main idea here is that (i) the “volume” is determined both by the “inter-atomic distance” and the “largest Pb-I-Pb angle” (which determines the size of the cavity for MA and is exactly the phonon coordinate of 1 THz mode), and (ii) the commonly observed temperature dependent decrease of the band gap in the conventional semiconductors comes mainly from the “inter-atomic distance”. In this paper ([10.1126/sciadv.1601156]), the MD simulation has been performed without any constraint on the “largest Pb-I-Pb angle”, which means that the authors didn’t rule out the contribution of the gradual change of the averaged “largest Pb-I-Pb angle” (the averaged phonon coordinate of 1 THz phonon) by increasing temperature.

More importantly, what we observe experimentally without any interpretation is that even without “thermal expansion” from any other phonon contribution, the band gap was increased only by selectively shifting the averaged phonon coordinate of 1 THz mode.

Therefore, we added the following sentence in the introduction to mention this work,

“A recent theoretical study has also reported the positive correlation between thermal expansion and the band gap. However, underlying both the thermal expansion and band gap variations is caused by the complex, temperature-dependent behavior of phonons.”

and we explain the role of 1 THz phonon in increasing the band gap in terms of our experimental observation in the following paragraph:

“ ...

When the sample temperature increases, the population of all thermally accessible phonons increases and, in turn, the coordinates of these phonons are displaced. However, each phonon has a different potential contribution to the band gap shift by its population depending on its coupling with the band gap. In case of this 1 THz phonon, we were able to selectively increase the phonon population and simultaneously observe the band gap shift: The THz excitation with the peak field strength of 100 kV/cm on our sample induces a blue-shift of the optical band gap by ~ 0.3 meV at 295 K (at $t = 0$, see SI). Although the selective excitation of the 1 THz phonon inherently leads to a non-equilibrium situation, one can define an effective temperature increase of the 1 THz phonon ΔT_{ph} , which can be estimated to be ~ 0.2 K (for the details see SI). Within the assumption of the Boltzmann distribution within the 1 THz mode, the efficiency of the 1 THz phonon in increasing the band gap (the slope = $(\Delta E_{gap})/(\Delta T_{ph}) \sim 1.5$ meV K^{-1}) is larger than that of overall thermally accessible phonon (the slope = $(\Delta E_{gap})/(\Delta T_{actual}) \sim 0.1$

meV K⁻¹ in the measured sample, Figure 5). This qualitative comparison of these slopes indicates that the thermal population of the 1 THz phonon is expected to have a significant contribution to the temperature dependence of the band gap, compensating potentially for other phonon mode population or lattice effects.”

5) The discussion lacks any reference to excitons. The authors are probing optical transitions right at the edge of the first bound exciton, but no discussion is provided on the effect of electron-hole interactions on their measurements, interpretation, and conclusions.

→ Thank you for your comment. Accordingly, we added the following sentence in the main text:

“Even though the optical transition near the band gap in semiconductors can contain contributions from the exciton, the exciton binding energy in this perovskite has been reported to be a few to a few tens of meV which makes the contribution from the exciton negligible at room temperature.”

6) The Franck-Condon diagram sketched in Fig. 3 is not relevant and misleading. The electronic structure of the inorganic sublattice of MAPbI₃ cannot be compared to a molecular system, in fact it is much closer to a III-V semiconductor like GaAs. In particular the potential energy surfaces in the ground state and in the excited state of MAPbI₃ differ only by a rigid shift and do not have distinct minima. In fact the exciton binding energies are very small (10-30 meV depending on the measurement) hence this system resides in the Wannier regime, not the Frenkel regime as Fig. 3 would imply.

→ Thank you for your concern. We agree that if a schematic diagram could be (unintentionally) misleading, and so indeed perhaps it is better not to show it. In addition, the potential curve is not proven to have this actual shape. Accordingly, we have removed this figure completely.

Reviewer #3 (Remarks to the Author):

The manuscript describes a bandgap bleaching features with oscillation of ~ 1 THz by ultrafast near-resonant THz phonon excitation in a perovskite semiconductor. With the help of simulations, authors explains the temperature dependence of the gap in the tetragonal phase as the thermal population of the 1 THz phonon mode is the major cause of thermal-shift in the bandgap. The experiment is well designed and results are intriguing. However, there are a few issues and concerns below that should be adequately addressed.

(1) The coherent phonons oscillation seen in the white light spectra is very convincing to conclude that there is a strong coupling of the lattice and electronic degrees of freedom. However, it is of much more speculative to further extend this to the coupling between a specific phonon and optical properties. The authors provide arguments mostly based on indirect evidence such as estimations of temperatures increase and subtle phase difference. However, such a claim really calls a clear cut experiment feature, e.g., if authors can shift excitation spectra to 2THz Does it give the same effect to the band gap? It is probably

technically difficult to do that. Anyway a more convincing data is needed to support the mode-specific nature. There are more problems in the data for such a claim, e.g., temperature dependence of modulation frequency changes from 2THz to ~ 1.8 THz, does this correlate with the temperatures dependence shift of phonons? I suggest authors improve this and further justify the claim which can be a substantial progress.

→ Thank you for the suggestion. Following the reviewer's suggestion, we have repeated the experiments with different spectral contents of the terahertz pump pulse while retaining the peak field strength. We observe exactly the same effect, with a broadened terahertz pulse. We have included these new results in Figure S4 in the supplementary information. As pointed out by the reviewer, totally shifting the THz spectra towards 2 THz is indeed technically difficult (due to the conversion efficiency of the THz generating crystal).

The temperature dependence of the modulation frequency changing from 2 to 1.8 THz is indeed fully compatible with the temperature dependence shift of the phonon frequency. We agree that the correlation between temperature dependent modulation frequency of the bandgap and the temperature dependent phonon frequency is a very important point, and we have included an additional figure (Figure 4) in the main text. Accordingly, we discuss this behavior in detail in the revised paragraph.

“Next, we investigate this specific phonon-gap coupling at several other temperatures, since the properties of both the band gap and the phonon depend on temperature. For all studied temperatures (81, 185, 225, 265, 295 K), the monotonically decaying and oscillating contributions are observed as shown in Figure 4a (and Figure S8). For the highest three temperatures, the terahertz-induced bandgap shift is very similar to that at 185K (see Figure S9), with the inferred central frequency and lifetime of the ~ 1 THz phonon exhibiting a temperature dependence in agreement with results from previous linear, steady-state spectroscopy (Figures 4b and 4c). All transient spectra ($t > 1$ ps) at different temperatures in the tetragonal phase can be accurately represented by a similar blue shift of the linear absorption spectrum (Figure S9). For the measurement at 81 K, one has to realize that MAPbI₃ undergoes a phase transition from the tetragonal phase (at room temperature) to the orthorhombic phase below ~ 160 K, which is accompanied by a band gap jump and also by a splitting of the 1 THz phonon mode into two phonon modes at 0.75 and 0.95 THz. While both phonons are excited by the terahertz pulse, the phonon-induced modulation of the optical band gap occurs only at 1.5 THz, which reveals that only the 0.75 THz (and not the 0.95 THz phonon) mode is strongly coupled to the optical gap (see Figure 4b).”

(2) The modulation is highly damped in the time domain (< 4 ps) which produce a broaden width in the FFT spectra at least twice as large as compared to the linear THz absorption (~ 1 meV). This should be explained more.

→ Thank you for the comment. The damping time of the modulation is obtained from either the fit to the damped harmonic oscillator (Fig. 2c) or by analyzing the width of the ~ 2 THz peak (dip) in the Figure 1d as $\tau_{20} = 1/(2\pi\Delta_{20})$, where Δ_{20} is the width of the ~ 2 THz feature. We find $\tau_{20} = 0.78 \pm 0.14$ ps while the τ_{10} from width of the linear absorption at ~ 1 THz is 0.67 ps. We added the following paragraph discussing the dephasing (decoherence) time of both fundamental (from the linear THz absorption linewidth) and overtone (from our experiment) in the main text as follows:

“Having clarified the mechanism of each contribution to our time-dependent differential spectra, we examine the dynamic properties of this 1 THz phonon. The population relaxation time of the 1 THz phonon, τ_{11} is determined to be 1.7 ± 0.4 ps at 295 K, indicating the time scale of the excess energy redistribution (see SI). The dephasing time of the 1 THz phonon overtone, τ_{20} , which is directly obtained from the damping term of the oscillation, i.e. the width (Δ_{20}) of the peak (dip) at ~ 2 THz in Figure 1d ($\tau_{20}=1/(2\pi\Delta_{20})$), is 0.78 ± 0.14 ps at 295 K (Table S1). The overtone dephasing time of 1 THz phonon is comparable to the reported fundamental dephasing time $\tau_{10}\cong 0.67$ ps within the experimental error.”

(3) In several places of the text the authors seems to imply the ultrafast band gap shift as thermal effect “that the excess energy absorbed by specific excitation of the lattice phonon.” This is confusing to me. It seems the blue absorption decay correlates with the dephasing of the phonons, i.e., coherence time. This is inconsistent with the energy dissipation or thermal picture of the 1THz phonon. A related question: If the lattice is oscillating back and forth (or fully coherent), I would expect the near-band-gap absorption oscillate near zero. Why is there a finite blue shift in the band edge in the entire temporal profile?

→ Thank you for the concern. We agree with the reviewer that the explanation about the transient differential spectra was not sufficiently clear. We have revised the main text so that the two components - the transient blue-shift of the band gap (monotonically decaying differential spectra, reflecting population relaxation dynamics) and the periodic modulation of the band gap (oscillating differential spectra, reflecting overtone phonon coherence dephasing dynamics) – are more clearly distinguishable and understandable in terms of their origins.

“The monotonically decaying differential spectrum, which reflects a transient blue shift of the optical band gap, can be understood by the transient displacement of a specific lattice coordinate. Population of a vibrationally excited state $|1\rangle$ displaces the lattice coordinate, e.g. the Pb-I-Pb angle in case of ~ 1 THz phonon mode (for the assignment, see the next paragraph). Then, the displacement alters the probability of electronic transitions from state $|1\rangle$ to state $|e\rangle$ (i.e. any electronic excited state) as compared to that of the $|0\rangle \rightarrow |e\rangle$ transition. Thus, the reduced optical absorption near the band gap upon phonon excitation (Fig. 1b) indicates a strong coupling between the specific phonon and the band gap (i.e. the electronic states which determine the band gap). Here, we note that all transient spectra decay to zero at $t > 4$ ps, i.e. the optical band gap returns to its original state (Fig. 1b) without further spectral changes. This means that even though the energy absorbed by the specific phonon (see below) is redistributed over all other available phonon modes (through phonon-phonon coupling), the resulting population of other phonon modes has negligible effect on the electronic transitions near the band gap.

In order to identify the specific phonon causing the observed optical band gap shift, we investigated the oscillating behavior of the transient differential spectra at $t > 1$ ps. Fourier transformation of the time traces (Fig. 1c and d) provides the frequency of the oscillatory signal and its phase relative to the distinct instantaneous electro-absorption (around $t = 0$, Fig. 1c). By comparing the frequency-domain response recorded at 1.59 eV (red curve in Fig. 1c), where only the electro-absorption is present, to the signals at 1.61 and 1.62 eV, where all three contributions are present, we find that the oscillatory component results in a narrow peak (dip at 1.62 eV), at ~ 2 THz (Fig. 1d). As discussed above, two THz-sample interactions (Fig. 2c) lead to a wave packet oscillating at the overtone frequency of the phonon (i.e.

overtone phonon coherence). Therefore, the oscillating behavior of the differential spectra at ~ 2 THz indicates such a wave packet generation of a phonon centered at ~ 1 THz. In turn, the lattice oscillation along this phonon coordinate shifts the coupled optical band gap periodically."

(4) Figure s1: why is the absorbance still very high below the band gap of ~ 1.6 eV?

→ It is apparently a common artifact from the surface coverage (this is not because of the high absorbance, but of the scattering loss of the transmitted beam). This has been reported in [J. Phys. Chem. Lett. 2015, 6, 3466–3470]. We have clarified this in the revised manuscript, by including the statement:

"The linear optical absorption spectrum is shown in Figure 1a. We note that the high background below the band gap (Fig. 1a) is due to the surface morphology of the sample; this effect is later cancelled by obtaining differential spectra between with and without the THz pump."

(5) Fig S6 and S7 – these two data sets are very interesting and relevant for the conclusion if explained well. I don't understand why authors don't put them in the main text. What is the origin of the phonon splitting at low temperatures? Are these two modes from the same 1THz phonon or they are separate from different origins? Are there any theory to calculate on their effects on the band structure?

→ Thank you for your kind suggestion. We have indeed put these data in the figure 4 and have added an explanation in the following paragraph that has been included in the revised manuscript:

"Next, we investigate this specific phonon-gap coupling at several other temperatures, since the properties of both the band gap and the phonon depend on temperature. For all studied temperatures (81, 185, 225, 265, 295 K), the monotonically decaying and oscillating contributions are observed as shown in Figure 4a (and Figure S8). For the highest three temperatures, the terahertz-induced bandgap shift is very similar to that at 185K (see Figure S9), with the inferred central frequency and lifetime of the ~ 1 THz phonon exhibiting a temperature dependence in agreement with results from previous linear, steady-state spectroscopy (Figures 4b and 4c). All transient spectra ($t > 1$ ps) at different temperatures in the tetragonal phase can be accurately represented by a similar blue shift of the linear absorption spectrum (Figure S9). For the measurement at 81 K, one has to realize that MAPbI₃ undergoes a phase transition from the tetragonal phase (at room temperature) to the orthorhombic phase below ~ 160 K, which is accompanied by a band gap jump and also by a splitting of the 1 THz phonon mode into two phonon modes at 0.75 and 0.95 THz. While both phonons are excited by the terahertz pulse, the phonon-induced modulation of the optical band gap occurs only at 1.5 THz, which reveals that only the 0.75 THz (and not the 0.95 THz phonon) mode is strongly coupled to the optical gap (see Figure 4b). The transient blue shift of the band gap by the population of this specific phonon at each temperature in the tetragonal phase well explains the transient differential spectra at $t > 0$ (Figure S9)."

Regarding the phonon splitting at low temperature, the paper that reports this phenomenon is [J. Phys. Chem. Lett. 2016, 7, 1–6]. The authors actually found the peak splitting not only for the 1 THz mode, but

also for the 2 THz mode. The origin of the phonon splitting, according to their explanation, is the significant change in the lattice parameter ($c(\text{tet.}) = 886 \text{ pm}$ and $c(\text{orth.}) = 856 \text{ pm}$) which is along the b axis. The split vibrational modes disappear when the symmetry along the a and b axes is lifted up. We further quote from their paper: "A similar explanation of the mode splitting due to the tetragonal-to-orthorhombic structural phase transition was given in an iron pnictide superconductor system ([Phys. Rev. B: Condens. Matter Mater. Phys. 2009, 80, 094504])." However, we were not able to find any theoretical work to calculate on their effects on the band structure in this material.

In summery, this is a potentially interesting paper and target relevant questions for the community. However, there are some substantial issues, as listed above, which should be clarified.

Reviewers' comments:

Reviewer #2 (Remarks to the Author):

I commend the authors for addressing all the comments raised by myself and the other referees. I consider that the manuscript has improved substantially, however the authors are still exaggerating some of their claims:

In their rebuttal they agree with all my comments, and even admit that their previous estimate of the electron-phonon coupling strength was rather inaccurate. The revised coupling strength that they now find is 15 times larger than what is needed to explain the temperature dependence of the band gap. By using this value one obtains a band gap variation of 0.6 eV across the tetragonal phase of MAPbI₃, which is obviously in strong disagreement with every experiment reported so far.

→ We thank the reviewer very much for the encouraging words. The reviewer's comments were indeed helpful for us to revisit several aspects of our work. One more thing we find necessary to clarify is that the band gap shift by the population of the 1 THz phonon is one of phonon contributions (both positive and negative) to the macroscopic temperature dependence of the band gap (0.1 meV/K). In other words, although the effect of only 1 THz phonon is estimated from our work to give rise to a 1.5 meV increase of the band gap following a temperature increase of 1 K, the expected negative contributions (from other phonon populations or lattice effects) likely compensate this effect, leading to the overall change of the band gap to be 'only' 0.1 meV/K. Accordingly, we have modified the corresponding statement as follows:

"We note that the potential negative contributions (i.e. the decrease of the band gap by increasing other phonon populations or lattice effects) are also expected, as commonly observed in semiconductors.^{23,24} Therefore, this qualitative comparison of these slopes indicates that the thermal population of the 1 THz phonon has a considerable contribution to the temperature dependence of the band gap, compensating for other phonon mode population or lattice effects."

The authors address this point in their rebuttal, but they seem unwilling to accordingly tone down their claims in the manuscript. Quoting from the abstract and from the conclusions:

"Excitation of the 1 THz phonon causes a blue shift of the band gap, explaining the temperature dependence of the gap in the tetragonal phase."

"Hence, our results provide a rationale for the peculiar temperature dependence of the band gap in MAPbI₃."

These statements are in contradiction with the author's rebuttal, and are not supported by this study. There is no doubt that this is an interesting work, but I strongly recommend that the authors revisit their statements in the manuscript in order to offer a fairer interpretation and to avoid unnecessary and misleading claims.

→ We understand the reviewer's concern. We agree with the reviewer that although isolating the mode-specific contribution of 1 THz phonon to the band gap shift and the efficiency itself are significant, the 1 THz mode does not provide the sole/quantitative explanation for the phenomenon. Therefore, we further tone down the statements as follows:

“Excitation of the 1 THz phonon causes a blue shift of the band gap, favorably contributing to the temperature dependence of the gap in the tetragonal phase.”

“Hence, our results demonstrate a mode-resolved approach to understanding the peculiar temperature dependence of the band gap in MAPbI₃.”

Reviewer #3 (Remarks to the Author):

The authors have answered most of my questions satisfactorily. The revised manuscript is also significantly improved. There are two remaining issues regarding my prior comments: (1) selective pumping to match the phonon spectra. The authors answered this with figure S4. This is confusing. If one detune the pump away from 1 THz phonon, the effect should be diminished. Did author try this, e.g., filter the pulse to 0.5 THz or 1.5 THz? (2) If the population of the 1 THz mode causes the band shift, then one will expect a rise time instead of decay in Fig. 2b, since the population gets larger when the coherence is dephased, which in turn case bigger band shift. It will be good to clarify this. After these remaining issues clarified, this work should be published.

→ We thank the reviewer very much for the encouraging comments and suggestions. Regarding the first comment (1), we apologize that our revised Figure S4 could indeed be confusing. The point of changing the bandwidth, was to investigate how the response depends on the different terahertz components contained within the excitation pulse. In the originally included figure S4, we scaled the red curve of the THz pump spectra to highlight the difference between the bandwidths of black and red curves. However, presenting normalized spectra is indeed potentially confusing and it could wrongly be concluded that detuning the pump away from 1 THz still generates the same effect. Therefore, we replace the scaled-red curve to the unscaled (red) curve so that the spectral components of the two different THz pulses can be directly compared (*highlighted in SI*). The unscaled curves reflect that the intensity around 1 THz is more or less identical between the two pulses, explaining the oscillatory components of the signals are the same. We unfortunately do not have such narrow band filters (0.5 or 1.5 THz), but from the current direct comparison in the figure S4 (f), we believe it is now clearer that our spectral shape-dependent measurement is not very different from what the reviewer suggests (detuning to 0.5 THz or 1.5 THz). Here (in figure S4, we have significantly different spectral components around ~0.6 THz (S4f, between black and red), but both the exponential and oscillatory signal after t=1 ps are essentially identical for both the black and red curves (S4a-e).

For the second comment (2), we agree with the reviewer that if there is any source of population increase after t = 1, we should have observed the rise of the signal. However, the coherence that we observe is created between the ground (v=0) and second excited (v=2) states via two field-matter interactions (perturbation), and by losing the quantum coherence, the system returns back to the state before the perturbation (i.e. v=0). Accordingly, we modified the corresponding statements as follows:

“The dephasing time of the 1 THz phonon overtone, τ_{20} , which indicates the time scale of recovering the original state from the coherence between two states (i.e. states $|0\rangle$ and $|2\rangle$), is 0.78 ± 0.14 ps at 295 K (Table S1). We obtained the value from the damping term of the oscillation, i.e. the width (Δ_{20}) of the peak (dip) at ~ 2 THz in Figure 1d ($\tau_{20}=1/(2\pi\Delta_{20})$).”

Reviewers' comments:

Reviewer #3 (Remarks to the Author):

I found the authors' answer to my second question of phonon population vs band shift is quite speculative and confusing. I don't understand how one can talk about the temperature if the decayed state has no population change in the reservoir. This is important to be clarified since it also relates to the other referee's question on the exact nature of band shift vs phonon temperature. I have no problem for the quality of the data but the paper can be benefited for more clarifications on these issues.

→ First of all, we apologize that there was a miscommunication between the reviewer's question and our answer. We interpreted the previous question to be whether the coherent superposition between $v=0$ and $v=2$ is converted directly to the population state $v=1$ of the particular 1 THz phonon by dephasing process, so the answer was no. However, we now understand that the reviewer's question indicates 'population of all thermally accessible phonons' in more general way, in other words, heat/energy dissipation by dephasing of the coherence. Accordingly, we added more statements in discussion paragraph to avoid potential confusion as follows:

"The THz excitation of our sample with the peak field strength of 100 kV/cm induces a blue-shift of the optical band gap by ~ 0.3 meV at 295 K (at $t = 0$, see Supplementary Figure 1 and Methods section). The absorbed pump pulse energy is equivalent to the amount of heat that can increase the entire lattice temperature by $\Delta T_{avg} \sim 5 \times 10^{-3}$ K (Supplementary Equation 4), when both population of 1 THz phonon decays and the overtone dephases at longer times ($t > 4$ ps). The average temperature increase by energy dissipation (ΔT_{avg}) has no noticeable contribution to the band gap shift as shown in Fig. 1b at $t > 4$ ps. This implies that other thermally accessible phonons are not strongly coupled to the band gap (compared to the 1 THz mode). When this energy is specifically absorbed by the 1 THz mode by resonant excitation, one can define an effective temperature increase of the 1 THz phonon ΔT_{ph} . The phonon temperature ΔT_{ph} , which identifies the population increase of only the 1 THz phonon, is estimated to be ~ 0.04 K (for the details see Methods section)."

, in addition to our initial statements about heat dissipation:

"we note that all transient spectra decay to zero at $t > 4$ ps, i.e. the optical band gap returns to its original state (Fig. 1b) without further spectral changes. This means that even though the energy absorbed by the specific phonon (see below) is redistributed over all other available phonon modes (through phonon-phonon coupling),^{37,38} the resulting population of other phonon modes has negligible effect on the electronic transitions near the band gap."

Regarding the reviewer's concern about still discussing temperature, we would like to emphasize that there are two outcomes by the two THz-phonon interactions (i) phonon population to $v=1$ state and (ii) overtone coherence between $v=0$ and $v=2$ states. The THz pulse not only generates coherence, but also increases the population ($v=1$) of only 1 THz phonon resonantly and instantaneously. When we correlate the band gap shift with the population, we directly compare the 'instantaneous' population increase (of

only 1 THz mode) to the 'instantaneous' blue shift of the band gap (transient absorption spectral amplitude). As mentioned above, although energy dissipated from the phonon coherence can also contribute to the population increase (by $\sim 5 \times 10^{-3}$ K, which potentially contributes to the rise of the signal as the reviewer indicated), the contribution is almost negligible and this population increase does not occur exclusively to only 1 THz mode, but also all thermally accessible phonon modes. Therefore, in order to isolate and discuss the only effect from 1 THz phonon on the band gap shift, we focused on the 'resonant and instantaneous' population increase only of 1 THz phonon.

Regarding the validity of the phonon assignment, since both the monotonic and oscillating signal depend quadratically on the THz field strength and the amplitude of monotonic and oscillating signal proportional to the same transition dipole moment (and transition dipole moments of between 01 and 12 are also proportional to each other), one could rule out the possibility that an unknown phonon (apart from the 1 THz mode) only contributes to the monotonic signal but not to the oscillating signal, unless the particular phonon dephasing time is unrealistically shorter than the population lifetime. Therefore, we can reasonably say that the monotonically decreasing component is dominated by the 1 THz phonon population dynamics. There is also experimental support that at least phonons of center frequency below 1 THz have no observable contribution to the population, which actually the reviewer suggested to add, and we added this statement as follows:

"Here, the identical transient signals at $t > 1$ ps induced by THz pump pulses with a narrower bandwidth (Supplementary Figure 2) indicate that phonon modes with a center frequency below ~ 1 THz have no observable contribution to the transient blue shift."